# INDUCTIVE MATRIX COMPLETION BASED ON GRAPH NEURAL NETWORKS

**Muhan Zhang\***
Washington University in St. Louis
`muhan@wustl.edu`
*Now at Facebook

**Yixin Chen**
Washington University in St. Louis
`chen@cse.wustl.edu`

## ABSTRACT

We propose an inductive matrix completion model without using side information. By factorizing the (rating) matrix into the product of low-dimensional latent embeddings of rows (users) and columns (items), a majority of existing matrix completion methods are *transductive*, since the learned embeddings cannot generalize to unseen rows/columns or to new matrices. To make matrix completion *inductive*, most previous works use content (side information), such as user's age or movie's genre, to make predictions. However, high-quality content is not always available, and can be hard to extract. Under the extreme setting where not any side information is available other than the matrix to complete, can we still learn an inductive matrix completion model? In this paper, we propose an Inductive Graph-based Matrix Completion (IGMC) model to address this problem. IGMC trains a graph neural network (GNN) based purely on 1-hop subgraphs around (user, item) pairs generated from the rating matrix and maps these subgraphs to their corresponding ratings. It achieves highly competitive performance with state-of-the-art transductive baselines. In addition, IGMC is inductive – it can generalize to users/items unseen during the training (given that their interactions exist), and can even transfer to new tasks. Our transfer learning experiments show that a model trained out of the MovieLens dataset can be directly used to predict Douban movie ratings with surprisingly good performance. Our work demonstrates that: 1) it is possible to train inductive matrix completion models without using side information while achieving similar or better performances than state-of-the-art transductive methods; 2) local graph patterns around a (user, item) pair are effective predictors of the rating this user gives to the item; and 3) Long-range dependencies might not be necessary for modeling recommender systems.

## 1 INTRODUCTION

Matrix completion (Candès & Recht, 2009) is one common formulation of recommender systems, where rows and columns of a matrix represent users and items, respectively, and predicting users' interest in items corresponds to filling in the missing entries of the rating matrix. By assuming a low-rank rating matrix, many of the most popular matrix completion algorithms use factorization techniques that decompose a rating $r_{ij}$ into $\mathbf{w}_i^\top \mathbf{h}_j$, the inner product of user $i$'s and item $j$'s latent feature vectors $\mathbf{w}_i$ and $\mathbf{h}_j$, respectively, which have achieved great successes (Adomavicius & Tuzhilin, 2005; Schafer et al., 2007; Koren et al., 2009; Bobadilla et al., 2013)

However, matrix factorization is intrinsically transductive, meaning that the learned latent features (embeddings) for users/items are not generalizable to users/items unseen during the training. When the rating matrix has changed values or has new rows/columns added, it often requires a complete retraining to get the new embeddings. To make matrix completion inductive, Inductive Matrix Completion (IMC) has been proposed, which leverages content (side information) of users and items (Jain & Dhillon, 2013; Xu et al., 2013). In IMC, a rating is decomposed by $r_{ij} = \mathbf{x}_i^\top \mathbf{Q} \mathbf{y}_j$, where $\mathbf{x}_i$ and $\mathbf{y}_j$ are content feature vectors of user $i$ and item $j$, respectively, and $\mathbf{Q}$ is a learnable matrix modeling the feature interactions. To accurately predict missing entries, IMC methods have strong constraints on the content quality, which often leads to inferior performance when high-quality content is not available. Other content-based recommender systems (Lops et al., 2011) face similar problems. In

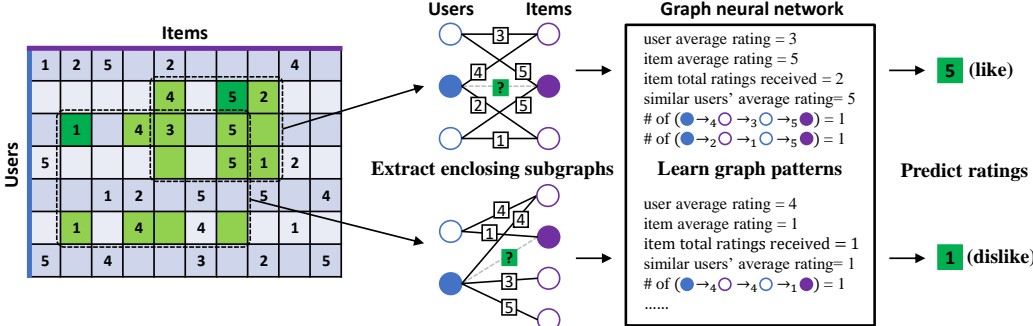

Figure 1: We extract a local enclosing subgraph around each rating (dark green), and train a GNN to map subgraphs to ratings. Each enclosing subgraph is induced by the user and item associated with the target rating as well as their $h$-hop neighbors (here $h=1$). From the subgraphs, a GNN can learn mixed graph patterns (such as average ratings, paths, etc.) useful for rating prediction. We illustrate some possible patterns in the GNN box. We use the trained GNN to complete the missing entries.

some extreme settings, there is not any content available for user, such as a website where users are completely anonymous. In these cases, inductive matrix completion seems impossible.

Recently, Hartford et al. (2018) propose exchangeable matrix layers, which apply permutation equivariant operations on the rating matrix to reconstruct missing entries. The resulting models are inductive and do not rely on side information. However, these models take the whole rating matrix **R** as input and output another reconstructed matrix, which poses scalability challenges for practical datasets with millions of users/items. In this paper, we propose a novel inductive matrix completion method that does not use any content. Further, our method does not need to take the whole rating matrix as input, and can infer ratings for individual user-item pairs. The key that frees us from using content or whole rating matrix is **local graph pattern**. If for each observed rating we add an edge between the corresponding user and item, we can build a *bipartite graph* from the rating matrix. Subsequently, predicting unknown ratings converts equivalently to predicting labeled links in this bipartite graph. This transforms matrix completion into a link prediction problem (Liben-Nowell & Kleinberg, 2007), where graph patterns play a major role in determining link existences.

A major class of link prediction methods are heuristic methods, which predict links based on some heuristic scores. For example, the common neighbors heuristic count the common neighbors between two nodes to predict links, while the Katz index (Katz, 1953) uses a weighted sum of all the walks between two nodes. See (Liben-Nowell & Kleinberg, 2007) for an overview. These heuristics can be seen as some predefined graph structure features calculated based on the local or global graph patterns around links, which have achieved great successes due to their simplicity and effectiveness.

However, these traditional link prediction heuristics only work for simple graphs where nodes and edges both only have a single type. Can we find some heuristics for labeled link prediction in bipartite graph? Intuitively, such heuristics should exist. For example, if a user $u_0$ likes an item $v_0$, we may expect to see very often that $v_0$ is also liked by some other user $u_1$ who shares a similar taste to $u_0$. By similar taste, we mean $u_1$ and $u_0$ have together both liked some other item $v_1$. In the bipartite graph, such a pattern is realized as a "like" path ($u_0 \rightarrow_{\text{like}} v_1 \rightarrow_{\text{liked by}} u_1 \rightarrow_{\text{like}} v_0$). If there are many such paths between $u_0$ and $v_0$, we may infer that $u_0$ is highly likely to like $v_0$. Thus, we may count the number of such paths as an indicator of how likely $u_0$ likes $v_0$. In fact, many neighborhood-based recommender systems (Desrosiers & Karypis, 2011) rely on similar heuristics.

Of course we can try to manually define many such intuitive heuristics and test their effectiveness. In this work, however, we take a different approach that **automatically learns suitable heuristics** from the given bipartite graph. To do so, we first extract an *$h$-hop enclosing subgraph* for each training user-item pair $(u, v)$, which is defined to be the subgraph induced from the bipartite graph by nodes $u, v$ and their neighbors within $h$ hops. Such local subgraphs contain rich graph pattern information about the rating that $u$ may give to $v$. For example, all the ($u_0 \rightarrow_{\text{like}} v_1 \rightarrow_{\text{liked by}} u_1 \rightarrow_{\text{like}} v_0$) paths are included in the 1-hop enclosing subgraph around $(u_0, v_0)$. By feeding these enclosing subgraphs to a graph neural network (GNN), we train a graph regression model that maps each subgraph to the rating that its center user gives to its center item. Figure 1 illustrates the overall framework.

Due to the superior graph learning ability, a GNN can learn highly expressive graph structure features useful for inferring the ratings without restricting the features to predefined heuristics. Given a trained GNN, we can also apply it to unseen users/items without retraining. Our resulting algorithm is inductive, and is named Inductive Graph-based Matrix Completion (IGMC). Note that IGMC **does not** address the extreme cold-start problem, as it still requires an unseen user-item pair's enclosing subgraph (i.e., the user and item should at least have some interactions with neighbors so that the enclosing subgraph is not empty). This scenario is very common in practice. For example, a newly registered YouTube user may quickly watch some videos without completing their personal information. In this case, if we cannot retrain user embeddings frequently, IGMC can be of great value by still making recommendations based purely on this user's interaction history with videos.

We compare IGMC with state-of-the-art matrix completion algorithms on five benchmark datasets. Without using any content, IGMC achieves the smallest RMSEs on four of them, even beating many transductive baselines augmented by side information. Our model is also equipped with excellent transfer learning ability. We show that an IGMC model trained on the MovieLens-100K dataset can be directly used to predict Douban movie ratings and even outperforms some baselines trained specifically on Douban. We also analyze IGMC's behavior on sparse rating matrices. We show that IGMC is more robust than transductive methods on sparse matrices. Under an extremely sparse case (only 0.1% of MovieLens-1M training ratings are kept), IGMC can still achieve less than 0.95 RMSE, beating a state-of-the-art transductive method GC-MC by more than 0.1 RMSE. Finally, our visualization confirms that local enclosing subgraphs are indeed strong predictors of ratings.

## 2 RELATED WORK

**Graph neural networks** Graph neural networks (GNNs) are a new type of neural networks for learning over graphs (Scarselli et al., 2009; Bruna et al., 2013; Duvenaud et al., 2015; Li et al., 2015; Kipf & Welling, 2016; Niepert et al., 2016; Dai et al., 2016). There are two types of GNNs: **Node-level GNNs** use message passing layers to iteratively pass messages between each node and its neighbors in order to extract a feature vector for each node encoding its local substructure. **Graph-level GNNs** additionally use a pooling layer such as summing which aggregates node feature vectors into a graph representation to enable graph-level tasks such as graph classification/regression. Due to the superior graph representation learning ability, GNNs have achieved state-of-the-art performance on semi-supervised node classification (Kipf & Welling, 2016), network embedding (Hamilton et al., 2017), graph classification (Zhang et al., 2018), and link prediction (Zhang & Chen, 2018), etc.

**GNNs for matrix completion** The matrix completion problem has been studied using GNNs. Monti et al. (2017) develop a multi-graph CNN (MGCNN) model to extract user and item latent features from their respective nearest-neighbor networks. Berg et al. (2017) propose graph convolutional matrix completion (GC-MC) which directly applies a GNN to the user-item bipartite graph to extract user and item latent features using a GNN. The SpectralCF model of (Zheng et al., 2018) uses a spectral-GNN on the bipartite graph to learn node embeddings. Although using GNNs for matrix completion, all these models are still **transductive** – MGCNN and SpectralCF require graph Laplacians which do not generalize to new graphs, while GC-MC uses one-hot encoding of node IDs as initial node features, thus cannot generalize to unseen users/items. A recent inductive graph-based recommender system, PinSage (Ying et al., 2018a), uses node content as initial node features (instead of the one-hot encoding in GC-MC), and is successfully used in recommending related pins in Pinterest. Although being inductive, PinSage relies heavily on the rich visual and text content associated with the pins which is not often accessible in other recommendation tasks. In comparison, our IGMC model is inductive and does not rely on any content. All previous approaches use node-level GNNs to learn embeddings for **nodes**, while our IGMC uses a **graph-level** GNN to learn representations for **subgraphs**. We will discuss this crucial difference in more details in Section 4.

Another related previous work is (Hartford et al., 2018), which defines exchangeable matrix layers to perform permutation-equivariant operations on matrices to achieve inductive matrix completion without using content. In particular, the operation updates each matrix entry by a weighted sum of itself, entries of its row, entries of its column, and all other entries of the matrix, where parameters for each of the four components are shared across all entries. It can also be regarded as a GNN with the exceptions that 1) the message passing is performed on edges (final edge features are pooled into node features), and 2) all edges (including those not connected to the center edge) pass messages

---

**Algorithm 1** ENCLOSING SUBGRAPH EXTRACTION

---

1: **input:** $h$, target user-item pair $(u, v)$, the bipartite graph $G$
2: **output:** the $h$-hop enclosing subgraph $G_{u,v}^h$ for $(u, v)$
3: $U = U_{fringe} = \{u\}, V = V_{fringe} = \{v\}$
4: **for** $i = 1, 2, \ldots, h$ **do**
5:     $U'_{fringe} = \{u_i : u_i \sim V_{fringe}\} \setminus U$
6:     $V'_{fringe} = \{v_i : v_i \sim U_{fringe}\} \setminus V$
7:     $U_{fringe} = U'_{fringe}, \ V_{fringe} = V'_{fringe}$
8:     $U = U \cup U_{fringe}, \ V = V \cup V_{fringe}$
9:     Let $G_{u,v}^h$ be the vertex-induced subgraph from $G$ using vertices $U, V$
10:     Remove edge $(u, v)$ from $G_{u,v}^h$.
11: **end for**
12: **return** $G_{u,v}^h$
Note: $\{u_i : u_i \sim V_{fringe}\}$ is the set of nodes that are adjacent to at least one node in $V_{fringe}$ with any edge type.

---

to the center edge in each round. One limitation of (Hartford et al., 2018) is that it takes the entire rating matrix as input, which might raise concerns for large matrices. In comparison, our IGMC takes only local subgraphs as input which avoids the issue and enables predicting individual ratings.

**Link prediction based on graph patterns** Learning supervised heuristics (graph patterns) has been studied for link prediction in simple graphs. Zhang & Chen (2017) propose Weisfeiler-Lehman Neural Machine (WLNM), which learns graph structure features using a fully-connected neural network on the subgraphs' adjacency matrices. Later, they improve this work by replacing the fully-connected neural network with a GNN and achieves state-of-the-art link prediction results (Zhang & Chen, 2018). Our work generalizes this line of research from predicting link existence in simple graphs to predicting values of links in bipartite graphs (i.e., matrix completion). In (Chen et al., 2005; Zhou et al., 2007), traditional link prediction heuristics are adapted to bipartite graphs which show promising performance for recommender systems. Our work differs in that we do not use any predefined heuristics, but learn general graph structure features using a GNN. Another similar work to ours is (Li & Chen, 2013), where graph kernels are used to learn graph structure features. However, graph kernels require quadratic time and space complexity to compute and store the kernel matrices thus are unsuitable for modern recommender systems.

## 3 INDUCTIVE GRAPH-BASED MATRIX COMPLETION (IGMC)

We now present our Inductive Graph-based Matrix Completion (IGMC) framework. We use $G$ to denote the undirected bipartite graph constructed from the given rating matrix $\mathbf{R}$. In $G$, a node is either a user (denoted by $u$, corresponding to a row in $\mathbf{R}$) or an item (denoted by $v$, corresponding to a column in $\mathbf{R}$). Edges can exist between user and item, but cannot exist between two users or two items. Each edge $(u, v)$ has a value $r = \mathbf{R}_{u,v}$, corresponding to the rating that $u$ gives to $v$. We use $\mathcal{R}$ to denote the set of all possible ratings (e.g., $\mathcal{R} = \{1, 2, 3, 4, 5\}$ in MovieLens), and use $\mathcal{N}_r(u)$ to denote the set of $u$'s neighbors that connect to $u$ with edge type $r$.

### 3.1 ENCLOSING SUBGRAPH EXTRACTION

The first part of IGMC is enclosing subgraph extraction. For each observed rating $\mathbf{R}_{u,v}$, we extract an $h$-hop enclosing subgraph around $(u, v)$ from $G$. Algorithm 1 describes the BFS procedure for extracting $h$-hop enclosing subgraphs. We will feed these enclosing subgraphs to a GNN and regress on their ratings. Then, for each testing $(u, v)$ pair, we again extract its $h$-hop enclosing subgraph from $G$, and use the trained GNN model to predict its rating. Note that after extracting a training enclosing subgraph for $(u, v)$, we should remove the edge $(u, v)$ because it is the target to predict.

### 3.2 NODE LABELING

The second part of IGMC is node labeling. Before we feed an enclosing subgraph to the GNN, we first apply a *node labeling* to it, which gives an integer label to every node in the subgraph. The

purpose is to use different labels to mark nodes' different roles in a subgraph. Ideally, our node labeling should be able to: 1) distinguish the target user and target item between which the target rating is located, and 2) differentiate user-type nodes from item-type nodes. Otherwise, the GNN cannot tell between which user and item to predict the rating, and might lose node-type information. To satisfy these conditions, we propose a node labeling as follows: We first give label 0 and 1 to the target user and target item, respectively. Then, we determine other nodes' labels according to at which hop they are included in the subgraph in Algorithm 1. If a user-type node is included at the $i^{\text{th}}$ hop, we will give it a label $2i$. If an item-type node is included at the $i^{\text{th}}$ hop, we will give it $2i + 1$. Such a node labeling can sufficiently discriminate: 1) target nodes from "context" nodes, 2) users from items (users always have even labels), and 3) nodes of different distances to the target rating.

Note that this is not the only possible way of node labeling, but we empirically verified its excellent performance. The one-hot encoding of these node labels will be treated as the initial node features $\mathbf{x}^0$ of the subgraph when fed to the GNN. Note that our node labels are determined completely **inside each enclosing subgraph**, thus are independent of the global bipartite graph. Given a new enclosing subgraph, we can as well predict its rating even if all of its nodes are from a different bipartite graph, because IGMC purely relies on graph patterns within local enclosing subgraphs without leveraging any global information specific to the bipartite graph. Our node labeling is also **different** from using the **global node IDs** as in GC-MC (Berg et al., 2017). Using one-hot encoding of global IDs is essentially transforming the first message passing layer's parameters into latent node embedding associated with each particular ID (equivalent to an embedding lookup table). Such a model cannot generalize to nodes whose IDs are out of range, thus is transductive.

### 3.3 Graph neural network architecture

The third part of IGMC is to train a graph neural network (GNN) model predicting ratings from the enclosing subgraphs. In previous node-based approaches such as GC-MC, a node-level GNN is applied to the entire bipartite graph to extract node embeddings. Then, the node embeddings of $u$ and $v$ are input to an inner-product or bilinear operator to reconstruct the rating on $(u, v)$. In contrast, IGMC applies a graph-level GNN to the enclosing subgraph around $(u, v)$ and maps the subgraph to the rating. There are thus two components in our GNN: 1) message passing layers that extract a feature vector for each node in the subgraph, and 2) a pooling layer to summarize a subgraph representation from node features.

To learn the rich graph patterns introduced by the different edge types, we adopt the relational graph convolutional operator (R-GCN) (Schlichtkrull et al., 2018) as our GNN's message passing layers, which has the following form:

$$\mathbf{x}_i^{l+1} = \mathbf{W}_0^l \mathbf{x}_i^l + \sum_{r \in \mathcal{R}} \sum_{j \in \mathcal{N}_r(i)} \frac{1}{|\mathcal{N}_r(i)|} \mathbf{W}_r^l \mathbf{x}_j^l, \tag{1}$$

where $\mathbf{x}_i^l$ denotes node $i$'s feature vector at layer $l$, $\mathbf{W}_0^l$ and $\{\mathbf{W}_r^l | r \in \mathcal{R}\}$ are learnable parameter matrices. Since neighbors $j$ connected to $i$ with different edge types $r$ are processed by different parameter matrices $\mathbf{W}_r^l$, we are able to learn a large amount of enriched graph patterns inside the edge types, such as the average rating the target user gives to items, the average rating the target item receives, and by which paths the two target nodes are connected, etc. We stack $L$ message passing layers with *tanh* activations between two layers. Following (Zhang et al., 2018; Xu et al., 2018), node $i$'s feature vectors from different layers are concatenated as its final representation $\mathbf{h}_i$:

$$\mathbf{h}_i = \text{concat}(\mathbf{x}_i^1, \mathbf{x}_i^2, \ldots, \mathbf{x}_i^L). \tag{2}$$

Next, we pool the node representations into a graph-level feature vector. There are many choices such as summing, averaging, SortPooling (Zhang et al., 2018), DiffPooling (Ying et al., 2018b), etc. In this work, however, we use a different pooling layer which concatenates the final representations of only the target user and item as the graph representation:

$$\mathbf{g} = \text{concat}(\mathbf{h}_u, \mathbf{h}_v), \tag{3}$$

where we use $\mathbf{h}_u$ and $\mathbf{h}_v$ to denote the final representations of the target user and target item, respectively. Our particular choice is due to the extra importance that these two target nodes carry

compared to other context nodes. Although being very simple, we empirically verified its better performance than summing and other advanced pooling layers for our matrix completion tasks.

After getting the final graph representation, we use an MLP to output the predicted rating:

$$\hat{r} = \mathbf{w}^\top \sigma(\mathbf{W}\mathbf{g}), \tag{4}$$

where $\mathbf{W}$ and $\mathbf{w}$ are parameters of the MLP which map the graph representation $\mathbf{g}$ to a scalar rating $\hat{r}$, and $\sigma$ is an activation function (we take ReLU in this paper).

### 3.4 MODEL TRAINING

**Loss function** We minimize the mean squared error (MSE) between the predictions and the ground truth ratings:

$$\mathcal{L} = \frac{1}{|\{(u,v)|\mathbf{\Omega}_{u,v}=1\}|} \sum_{(u,v):\mathbf{\Omega}_{u,v}=1} (R_{u,v} - \hat{R}_{u,v})^2, \tag{5}$$

where we use $R_{u,v}$ and $\hat{R}_{u,v}$ to denote the true rating and predicted rating of $(u,v)$, repsectively, and $\mathbf{\Omega}$ is a 0/1 mask matrix indicating the observed entries of the rating matrix $\mathbf{R}$.

**Adjacent rating regularization** The R-GCN layer (1) used in our GNN has different parameters $\mathbf{W}_r$ for different rating types. One drawback here is that it fails to take the magnitude of ratings into consideration. For instance, a rating of 4 and a rating of 5 in MovieLens both indicate that the user likes the movie, while a rating of 1 indicates that the user does not like the movie. Ideally, we expect our model to be aware of the fact that a rating of 4 is more similar to 5 than 1 is. In R-GCN, however, ratings 1, 4 and 5 are only treated as three independent edge types – the magnitude and order information of the ratings is completely lost. To fix that, we propose an adjacent rating regularization (ARR) technique, which encourages ratings adjacent to each other to have similar parameter matrices. Assume the ratings in $\mathcal{R}$ exhibit an ordering $r_1, r_2, \ldots, r_{|\mathcal{R}|}$ which indicates increasingly higher preference that users have for items. Then, the ARR regularizer is:

$$\mathcal{L}_{\text{ARR}} = \sum_{i=1,2,\ldots,|\mathcal{R}|-1} \|\mathbf{W}_{r_{i+1}} - \mathbf{W}_{r_i}\|_F^2, \tag{6}$$

where $\|\cdot\|_F$ denotes the Frobenius norm of a matrix. The above regularizer restrains the parameter matrices of adjacent ratings from having too much differences, which not only takes into consideration of the ratings' order, but also helps the optimization of those infrequent ratings by transferring knowledge from their adjacent ratings. The final loss function is given by:

$$\mathcal{L}_{\text{final}} = \mathcal{L} + \lambda \mathcal{L}_{\text{ARR}}, \tag{7}$$

where $\lambda$ trades-off the importance of the MSE loss and the ARR regularizer. There are many other ways to model rating magnitude and order, which are left for future work.

## 4 GRAPH-LEVEL GNN VS. NODE-LEVEL GNN

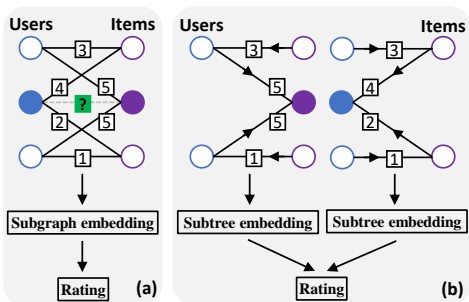

(a)

(b)

Compared to previous graph matrix completion approaches such as PinSage and GC-MC, one important difference of IGMC is that it uses a graph-level GNN to map the enclosing subgraph around the target user and item to their rating (left figure (a)), instead of using a node-level GNN on the bipartite graph $G$ to learn target user's and item's embeddings and use the node embeddings to predict the rating (left figure (b)). One drawback of the latter node-based approach is that the learned node embeddings are essentially encoding the two rooted subtrees around the two nodes **independently**, which fails to model the **interactions** and **correspondences** between the nodes of the two trees. For example, from the two subtrees of the left figure (b) we do not really know whether the two target nodes are just isolated from each other like in (b) or actually densely connected like in (a); these two cases look **identical** to a node-based approach.

Table 1: Statistics of each dataset.

| Dataset | Users | Items | Ratings | Density | Rating types |
|---------|-------|-------|---------|---------|--------------|
| Flixster | 3,000 | 3,000 | 26,173 | 0.0029 | 0.5, 1, 1.5, ..., 5 |
| Douban | 3,000 | 3,000 | 136,891 | 0.0152 | 1, 2, 3, 4, 5 |
| YahooMusic | 3,000 | 3,000 | 5,335 | 0.0006 | 1, 2, 3, ..., 100 |
| ML-100K | 943 | 1,682 | 100,000 | 0.0630 | 1, 2, 3, 4, 5 |
| ML-1M | 6,040 | 3,706 | 1,000,209 | 0.0447 | 1, 2, 3, 4, 5 |

In comparison, a graph-level GNN can discriminate the two cases through a sufficient number of message passing rounds. Since the learning is confined to the subgraph, stacking multiple graph convolution layers will learn more and more refined local structural features which can adequately discriminate up to all subgraphs that the Weisfeiler-Lehman algorithm can discriminate (Xu et al., 2018). However, for node-based approaches, since there is no subgraph boundary, stacking multiple graph convolutions will only extend the convolution range to unrelated distant nodes and over-smooth the node embeddings (Li et al., 2018). This is reflected in that previous node-based approaches mainly use only one or two message passing layers (Berg et al., 2017; Ying et al., 2018a).

Nevertheless, using a graph-level GNN on every target rating's enclosing subgraph has higher complexity than using a node-level GNN on the entire bipartite graph. Suppose the bipartite graph has $|E|$ edges. Then performing one round of message passing using a node-level GNN has $\mathcal{O}(|E|)$ complexity. Assume the maximum number of edges in all enclosing subgraphs is $K$. Performing one round of message passing for all enclosing subgraphs then has $\mathcal{O}(K|E|)$ complexity. In practice, we can use subsampling to restrict $K$ to a small number to reduce IGMC's complexity.

## 5 EXPERIMENTS

We conduct experiments on five common matrix completion datasets: Flixster (Jamali & Ester, 2010), Douban (Ma et al., 2011), YahooMusic (Dror et al., 2011), MovieLens-100K and MovieLens-1M (Miller et al., 2003). For ML-100K, we train and evaluate on the canonical u1.base/u1.test train/test split. For ML-1M, we randomly split it into 90% and 10% train/test sets. For Flixster, Douban and YahooMusic we use the preprocessed subsets and splits provided by (Monti et al., 2017). Dataset statistics are summarized in Table 1. We implemented IGMC using pytorch_geometric (Fey & Lenssen, 2019). We tuned model hyperparameters based on cross validation results on ML-100K, and used them across all datasets. The final architecture uses 4 R-GCN layers with 32, 32, 32, 32 hidden dimensions. Basis decomposition with 4 bases is used to reduce the number of parameters in $\mathbf{W}_r$ (Schlichtkrull et al., 2018). The final MLP has 128 hidden units and a dropout rate of 0.5. We use **1-hop** enclosing subgraphs for all datasets, and find them sufficiently good. We find using 2 or more hops can slightly increase the performance but take much longer training time. For each enclosing subgraph, we randomly drop out its adjacency matrix entries with a probability of 0.2 during the training. We set the $\lambda$ in (7) to 0.001. We train our model using the Adam optimizer (Kingma & Ba, 2014) with a batch size of 50 and an initial learning rate of 0.001, and multiply the learning rate by 0.1 every 20 epochs for ML-1M, and every 50 epochs for all other datasets. Our code is publicly available at `https://github.com/muhanzhang/IGMC`.

### 5.1 FLIXSTER, DOUBAN AND YAHOOMUSIC

For these three datasets, we compare our IGMC with GRALS (Rao et al., 2015), sRGCNN (Monti et al., 2017), GC-MC (Berg et al., 2017), F-EAE (Hartford et al., 2018), and PinSage (Ying et al., 2018a). Among them, GRALS is a graph regularized matrix completion algorithm. GC-MC and sRGCNN are transductive node-level-GNN-based matrix completion methods. F-EAE uses exchangeable matrix layers to perform inductive matrix completion without using content. PinSage is an inductive node-level-GNN-based model using content, which is originally used to predict related pins and is adapted to predicting ratings here. We further implemented an inductive GC-MC model (IGC-MC) which replaces the one-hot encoding of node IDs with the content features to make it

Table 2: RMSE test results on Flixster, Douban and YahooMusic.

| Model | Inductive | Content | Flixster | Douban | YahooMusic |
|---|---|---|---|---|---|
| GRALS | no | yes | 1.245 | 0.833 | 38.0 |
| sRGCNN | no | yes | 0.926 | 0.801 | 22.4 |
| GC-MC | no | yes | 0.917 | 0.734 | 20.5 |
| IGC-MC | yes | yes | 0.999±0.062 | 0.990±0.082 | 21.3±0.989 |
| F-EAE | yes | no | 0.908 | 0.738 | 20.0 |
| PinSage | yes | yes | 0.954±0.005 | 0.739±0.002 | 22.9±0.629 |
| IGMC (ours) | yes | no | **0.872**±0.001 | **0.721**±0.001 | **19.1**±0.138 |

Table 3: RMSE test results on MovieLens-100K (left) and MovieLens-1M (right).

| Model | Inductive | Content | ML-100K | Model | Inductive | Content | ML-1M |
|---|---|---|---|---|---|---|---|
| MC | no | no | 0.973 | PMF | no | no | 0.883 |
| IMC | no | yes | 1.653 | I-RBM | no | no | 0.854 |
| GMC | no | yes | 0.996 | NNMF | no | no | 0.843 |
| GRALS | no | yes | 0.945 | I-AutoRec | no | no | 0.831 |
| sRGCNN | no | yes | 0.929 | CF-NADE | no | no | **0.829** |
| GC-MC | no | yes | **0.905** | GC-MC | no | no | 0.832 |
| IGC-MC | yes | yes | 1.142 | IGC-MC | yes | yes | 1.259 |
| F-EAE | yes | no | 0.920 | F-EAE | yes | no | 0.860 |
| PinSage | yes | yes | 0.951 | PinSage | yes | yes | 0.906 |
| IGMC | yes | no | **0.905** | IGMC | yes | no | 0.857 |

inductive. The content in these datasets are presented in the form of user and item graphs. We summarize whether each algorithm is inductive and whether it uses content in Table 2.

We train our model for 40 epochs, and save the model parameters every 10 epochs. The final predictions are given by averaging the predictions from epochs 10, 20, 30 and 40. We repeat the experiment five times and report the average RMSEs. The baseline results are taken from (Hartford et al., 2018). Table 2 shows the results. Our model achieves the smallest RMSEs on all three datasets without using any content, significantly outperforming all the compared baselines, regardless of whether they are transductive or inductive. Further, except F-EAE, all the baselines have used content information to assist the matrix completion. This further highlights IGMC's great performance advantages without relying on content.

## 5.2 ML-100K AND ML-1M

We further conduct experiments on MovieLens datasets. Side information is present for both users (age, gender, occupation, etc.) and movies (genres). For ML-100K, we compare against matrix completion (MC) (Candès & Recht, 2009), inductive matrix completion (IMC) (Jain & Dhillon, 2013), geometric matrix completion (GMC) (Kalofolias et al., 2014), as well as GRALS, sRGCNN, GC-MC, F-EAE and PinSage. We train IGMC for 80 epochs and report the ensemble performance of epochs 50, 60, 70 and 80. For ML-1M, besides the baselines GC-MC, F-EAE and PinSage, we further include state-of-the-art algorithms including PMF (Mnih & Salakhutdinov, 2008), I-RBM (Salakhutdinov et al., 2007), NNMF (Dziugaite & Roy, 2015), I-AutoRec (Sedhain et al., 2015) and CF-NADE (Zheng et al., 2016). We train IGMC for 40 epochs and report the ensemble performance of epochs 25, 30, 35 and 40. The experiments are repeated five times and the average results are reported in Table 3 (standard deviations are less than 0.001). As we can see, IGMC achieves the best performance on ML-100K, in parallel with GC-MC despite that IGMC is an inductive model, while GC-MC is transductive and additionally uses content information. For ML-1M, IGMC cannot catch up with state-of-the-art transductive models such as CF-NADE and GC-MC, but outperforms other inductive models. We will analyze this dataset further in Section 5.3.

Table 4: RMSE of transferring the model trained on ML-100K to Flixster, Douban and YahooMusic.

| Model | Inductive | Content | Flixster | Douban | YahooMusic |
|-------|-----------|---------|----------|--------|------------|
| IGC-MC | yes | no | 1.290 | 1.144 | 25.7 |
| F-EAE | yes | no | 0.987 | 0.766 | 23.3 |
| IGMC (ours) | yes | no | **0.906** | **0.759** | **20.1** |

## 5.3 SPARSE RATING MATRIX ANALYSIS

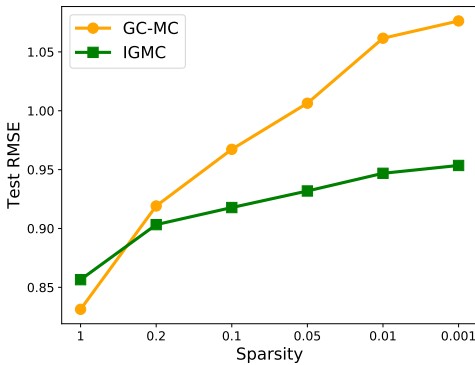

Figure 2: ML-1M results under different sparsity ratios.

To gain insight into when inductive graph-based matrix completion is more suitable than transductive methods, we compare IGMC with GC-MC on ML-1M under different sparsity levels of the rating matrix. We sequentially increase the sparsity level by randomly keeping only 0.2, 0.1, 0.05, 0.01, and 0.001 of the original training ratings. Then, we train both models on the sparsified rating matrices, and evaluate on the original test set. Figure 2 shows the results. As we can see, although IGMC falls behind GC-MC initially with full ratings, it starts to perform better after the sparsity ratio is less than 20%. The advantage becomes even greater under extremely sparse cases. This seems to indicate that IGMC is a better choice than transductive methods when there is not a large amount of training data, which is particularly suitable for the initial rating collection phase of a recommender system. It also suggests that transductive matrix completion relies more on the dense user-item interactions than inductive graph-based matrix completion does.

## 5.4 TRANSFER LEARNING

A great advantage of an inductive model is its potential for transferring to other tasks. We conduct a transfer learning experiment by applying the IGMC model trained on ML-100K to Flixster, Douban and YahooMusic. Among the three datasets, only Douban has exactly the same rating types as ML-100K (1,2,3,4,5). Thus for Flixster and YahooMusic, we bin their edge types into groups 1 to 5 before feeding into the ML-100K model, and multiply the YahooMusic predictions by 20 to account for the different scales. Despite all the compromises, the transferred IGMC model achieves excellent performance (Table 4). We also show the transfer learning results of other two inductive models, IGC-MC and F-EAE. Note that an inductive model using content features (such as PinSage) is not transferrable, due to the different feature spaces between MovieLens and the target datasets. Thus for IGC-MC, we replace its content features with node degrees. As we can see, IGMC outperforms the other two models by large margins in terms of transfer learning ability. Furthermore, the transferred IGMC even outperforms a wide range of baselines trained especially on each dataset (Table 2).

## 5.5 ABLATION STUDIES

To understand the individual contributions of some components in IGMC, we conduct several ablation studies. In particular, we are interested in: 1) whether the proposed pooling layer in Equation (3) helps; 2) whether the proposed adjacent rating regularization (ARR) helps; and 3) whether incorporating content can further improve IGMC's performance. We present the results in the appendix.

## 5.6 VISUALIZATION

Finally, we visualize 10 testing enclosing subgraphs with the highest and lowest predicted ratings for Flixster, Douban, YahooMusic, and ML-100K, respectively, in Figure 3. As we can see, there are substantially different patterns between high-score and low-score subgraphs, which is why IGMC can predict ratings merely from these subgraphs. For example, high-score subgraphs typically show

both high user average rating and high item average rating, while low-score subgraphs often have mixed ratings from non-target users and have low user average rating.

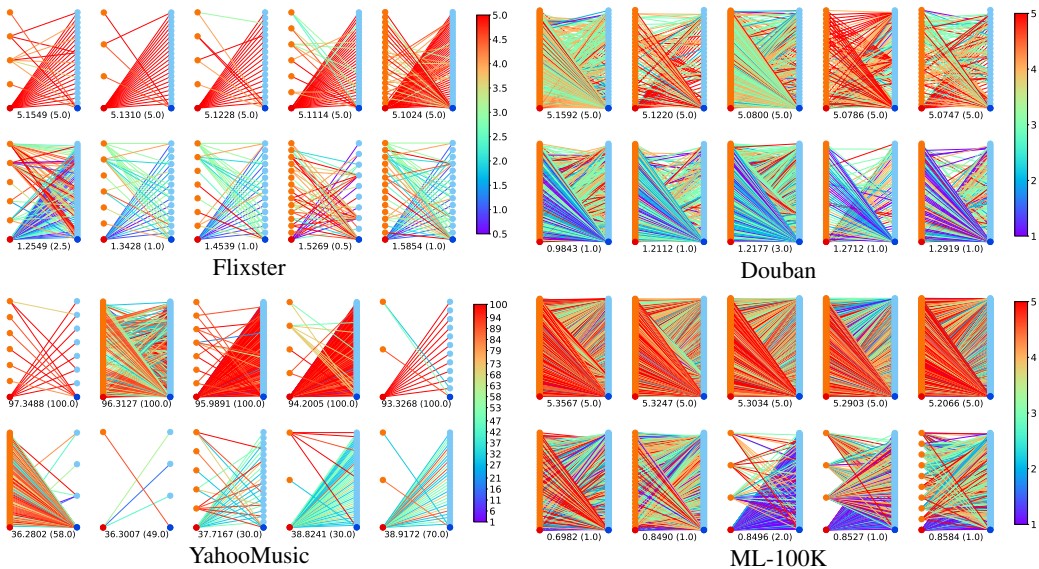

Figure 3: Testing enclosing subgraphs for Flixster, Douban, YahooMusic, and ML-100K. Top 5 and bottom 5 are subgraphs with the highest and lowest predicted ratings, respectively. In each subgraph, red nodes in the left are users, blue nodes in the right are items; the bottom red and blue nodes are the target user and item; the predicted rating and true rating (in parenthesis) are shown underneath. We visualize edge ratings using the color map shown in the right. Higher ratings are redder.

## 6  CONCLUSION

In this paper, we have proposed Inductive Graph-based Matrix Completion (IGMC). Instead of learning transductive latent features, IGMC learns local graph patterns related to ratings inductively based on graph neural networks. Compared to previous inductive matrix completion methods, IGMC does not rely on content (side information) of users/items. We show that IGMC has highly competitive performance compared to state-of-the-art baselines. In addition, IGMC is transferrable to new tasks without any retraining, a property much desired in those recommendation tasks having few training data. We hope IGMC can provide a new idea to matrix completion and recommender systems.

ACKNOWLEDGMENTS

The work is supported in part by the National Science Foundation under award numbers III-1526012 and SCH-1622678, and by the National Institute of Health under award number 1R21HS024581.

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

## A    ABLATION STUDIES

In this section, we conduct ablation experiments to study the individual contributions of some components in IGMC. In particular, we are interested in: 1) whether the proposed pooling layer in Equation (3) helps; 2) whether the proposed adjacent rating regularization (ARR) in Equation (6) helps; and 3) whether incorporating content can further improve IGMC's performance. Therefore, we compare the original IGMC with: 1) IGMC with SumPooling replacing the proposed pooling layer, 2) IGMC without ARR by setting $\lambda$ in (7) to 0, and 3) IGMC with content by concatenating target user and item's content feature vectors to the graph representation **g** before feeding into the MLP (4), which is similar to (Berg et al., 2017). The results are shown in Table 5.

Table 5: RMSE test results of ablation experiments.

| Model | Flixster | Douban | YahooMusic | ML-100K |
|---|---|---|---|---|
| IGMC (original) | **0.872**±0.001 | **0.721**±0.001 | **19.1**±0.138 | **0.905**±0.001 |
| IGMC (SumPooling) | 0.879±0.002 | 0.729±0.003 | 22.9±1.102 | 0.933±0.001 |
| IGMC (no_ARR) | **0.872**±0.001 | **0.721**±0.001 | 19.2±0.140 | 0.908±0.001 |
| IGMC (content) | 0.886±0.002 | 0.726±0.001 | **19.1**±0.069 | 0.906±0.001 |

From Table 5, we have the following observations. Firstly, using the proposed pooling layer shows a huge improvement over a standard SumPooling. This might be because SumPooling assigns equal importance to all nodes in a subgraph, which fails to distinguish the target user and item from the context nodes. This indicates that a pooling layer able to highlight the target user and item is important for IGMC.

Secondly, we can see that disabling ARR results in a 0.003 performance drop on ML-100K, while seeming to have no influence on the other three datasets. One possible explanation is that ARR is more useful for modeling large and dense enclosing subgraphs, since ARR enhances the modeling power by modeling the extra relationships between edge types in terms of their magnitude differences. Another possible reason is that we only tuned ARR's $\lambda$ on ML-100K and used $\lambda = 0.001$ uniformly on all datasets. This is partly verified by that when increasing $\lambda$ to 0.1 for YahooMusic, we can further decrease IGMC's RMSE to 18.98±0.140. We did not fully verify this, and leave better ways of modeling rating magnitude for future study.

Thirdly, we observe that incorporating content does not improve IGMC's performance on ML-100K, and often hurts the performance on the other datasets. For Flixster, Douban and YahooMusic, this phenomenon can be explained by that the content of these three datasets are user/item's respective graphs, which has little to no gains to a model that has already exploited graph structure information between users and items very well. In addition, the content feature vectors are presented in 3000-dimensional adjacency vectors, which might pose new problems due to their size and sparsity. For ML-100K, the lost of some performance when adding content actually contradicts with our initial experiments. In our initial experiments when the RMSE on ML-100K without content was around 0.910, we observed that adding content was indeed helpful and reduced the RMSE to

0.907. However, after we introduced ARR and redid the hyperparameter tuning, the RMSE of ML-100K without content became 0.905, and adding content no longer helped. We hypothesize that the benefits of content reduce with the better modeling of graph structure features.

The way we incorporate content might be another reason for why content is not useful in IGMC. In our experiments we only concatenate the target user and item's content vectors with the final graph representation output by the GNN, similar to (Berg et al., 2017). However, this method fails to model the interactions between content and graph structures in the early graph convolution stage. On the other hand, as Berg et al. (2017) and Zhang & Chen (2018) found, directly concatenating content with initial node features (the one-hot encoding vectors) as the input to GNN often led to worse performance due to information flow bottlenecks. We leave exploring better ways to combine content and graph structures for future work.

