# OpenReview forum: "Inductive Matrix Completion Based on Graph Neural Networks"
_ICLR.cc/2020/Conference — Accept (Spotlight)_

### Official Review · AnonReviewer2 · 2019-10-23
**Official Blind Review #2**

**Rating:** 6

**Review:**

This paper presents a method for inductive matrix completion that does not rely on side information to make predictions. The approach is as follows: select the sub-graph from an h-hop neighbourhood of any target user-item pair (u,v) and relabel it according to its distance from (u, v). The authors then treat rating prediction as graph classification problem given the selected subgraph and use a graph convolution architecture to make the prediction. They achieve strong empirical performance in both the transductive and inductive settings.

The paper presents an interesting approach to the inductive learning problem, with a number of interesting ideas and impressive empirical results. However, I have voted to reject this paper because I found the experimental section did a poor job of showing how the various ideas contribute to overall performance which made it difficult to evaluate the method beyond the absolute performance on benchmark datasets.

The key idea of the paper is interesting - samples h-hop enclosing subgraphs and label the resulting nodes by their hop-length and type (i.e. users at hop length h get a different label to items at length h).  In their experiments they find that a model with only a 1-hop neighbourhood to performs so well, which suggests that for recommender systems, supporting long chains of dependencies is unnecessary. This is a useful property of the data that the method depends on, because without it, h-hop neighbourhoods with larger h would quickly include the whole graph on most datasets (most real-world graphs have small diameter).

They process these subgraphs with relational graph convolutions, but make two modifications: first they build their final representations by concatenating all the message passing steps (eq. 2) and both user and item representations (eq. 3), and second they have adjacent rating regularization which constrains weights between adjacent ratings. Eq.2 in particular is unusual because it  amounts to a skip connection from every layer (somewhat analogous to a DenseNet [Huang et al. 2015]). The authors say both lead to improved performance - but don’t evaluate the approaches in the experimental section. What is the contribution of eq 2, 3 and 6 relative to more standard approaches (i.e. no skip connections, sum pooling, no regularization)?

Finally - I found the positioning of the work of the work misleading: from the abstract and related worked we're told that this is a "seemingly impossible problem" and prior work has relied on side information to deal with the inductive case. From this description, we might infer that this is the first approach that deals with the no-side information case. But the clear baseline is Hartford et al. 2018 which also doesn't use side information for inductive matrix completion; and yet it is not mentioned in either the introduction or the related work sections. This omission is unfortunate because both because it's misleading, and more importantly it misses the opportunity to discuss the relative merits of the approaches (beyond performance, where this paper provides a clear improvement). For example,
 - How do the two methods compare with respect to computational complexity? This approach seems more expensive because a different subgraph is selected for each prediction (i.e. 1 forward pass per test set point rather than a single forward pass to compute the full test set), but this isn't discussed.
 - Can you give any intuition about where the differences in performance are coming from?

------

Increased rating to Weak Accept following the rebuttal below.

**Experience Assessment:**

I have published one or two papers in this area.

**Review Assessment: Checking Correctness Of Derivations And Theory:**

N/A

**Review Assessment: Checking Correctness Of Experiments:**

I carefully checked the experiments.

**Review Assessment: Thoroughness In Paper Reading:**

I read the paper thoroughly.

---

> ### Author Response · Authors · 2019-11-11
> **Response to Reviewer 2**
>
> Thank you very much for your insightful and constructive comments.
>
> We have revised the paper based on them. In particular, we have made the following modifications:
> 1) We added some sentences to introduce [Hartford et al. 2018] in “Introduction” (page 2, first paragraph).
> 2) We added a paragraph in “Related work” that discusses [Hartford et al. 2018] and compares it to our work.
> 3) We added ablation experiments to study the individual contribution of each idea in Appendix A.
> 4)  Some small modifications to abstract and paper (such as removing the “seemingly impossible problem” claim).
>
> For your concrete comments:
> 1) Eq.2 in particular is unusual because it amounts to a skip connection from every layer (somewhat analogous to a DenseNet [Huang et al. 2015]). The authors say both lead to improved performance - but don’t evaluate the approaches in the experimental section.
> Concatenating each graph convolution layer’s output instead of only using the last layer is actually a common practice in GNN for graph classification [1,2,3]. The reason is to leverage multiple hops’ different graph structure features around each center node to achieve a maximum discriminative power that the Weisfeiler-Lehman algorithm has [2]. We didn’t claim it as our innovation.
>
> 2) What is the contribution of eq 2, 3 and 6 relative to more standard approaches (i.e. no skip connections, sum pooling, no regularization)?
> Thank you for suggesting the ablation studies. Please refer to Appendix A for the results.
>
> 3) How do the two methods compare with respect to computational complexity? This approach seems more expensive because a different subgraph is selected for each prediction (i.e. 1 forward pass per test set point rather than a single forward pass to compute the full test set), but this isn't discussed.
> Although IGMC requires 1 forward pass per test point, the forward pass is performed in batch form in our implementation by building a diagonal block adjacency matrix of multiple subgraphs. Suppose the maximum number of edges in a subgraph is restricted to K, then the complexity of one message passing in IGMC is O(K|E|), which is a constant times higher than a standard GCN message passing with O(|E|) complexity. In practice, we can set a small K and use subsampling to reduce computation time. In summary, [Harford et al. 2018] has lower complexity, predicts missing entries all together, while facing the large input matrix issues. IGMC has higher complexity, but is more flexible to choose which particular users and items to predict and has better performance.
>
> 4) Can you give any intuition about where the differences in performance are coming from?
> IGMC restricts the learning to the most useful local graph structures, rather than extending the convolution range to unrelated distant nodes or edges. Learning from local subgraphs makes IGMC’s GNN not waste the expressive power on faraway structures and focus on the most informative local ones. Furthermore, the subgraph boundary makes GNN able to learn refined graph patterns through multiple graph convolutions, achieving maximum discriminative power that the Weisfeiler-Lehman algorithm has. Without the boundary, GNN might be overwhelmed by the broad graph patterns in the global bipartite graph.
>
> [1] Zhang, M., Cui, Z., Neumann, M., & Chen, Y. (2018). An end-to-end deep learning architecture for graph classification. In Thirty-Second AAAI Conference on Artificial Intelligence.
> [2] Xu, K., Hu, W., Leskovec, J., & Jegelka, S. (2018). How powerful are graph neural networks? In ICLR-2019.
> [3] Xu, K., Li, C., Tian, Y., Sonobe, T., Kawarabayashi, K. I., & Jegelka, S. (2018). Representation Learning on Graphs with Jumping Knowledge Networks. In International Conference on Machine Learning (pp. 5449-5458).
> [Hartford et al. 2018] Hartford, J., Graham, D. R., Leyton-Brown, K., & Ravanbakhsh, S. (2018). Deep models of interactions across sets. arXiv preprint arXiv:1803.02879.

---

> > ### Comment · AnonReviewer2 · 2019-11-11
> > **response**
> >
> > Thanks for the detailed response - I've increased my score in response to your additions.

---

> > > ### Author Response · Authors · 2019-11-11
> > > **Thank you for raising the score**
> > >
> > > Thank you for the very helpful comments again! We really appreciate them.

---

### Official Review · AnonReviewer1 · 2019-10-24
**Official Blind Review #1**

**Rating:** 8

**Review:**

The paper uses a new approach for inductive matrix completion (IMC). The IMC problem is used to model recommender systems where there can be users/items with few (or even no) observed ratings, but where each user/item is associated with additional meta-data or other features. Since users/items close to each other in the feature space plausibly share similar ratings, one can inductively fill in the ratings even in the very sparse regime where traditional matrix completion fails.

The current paper demonstrates an interesting new approach for IMC that does not require any metadata about new users/items, but rather, exploits the structure of the observed ratings themselves. Specifically, the method (called Inductive Graph-based Matrix Completion, or IGMC) constructs a (fairly large) bipartite graph out of the observed user-item pairs, and creates subgraphs by looking at h-hop neighborhoods of each test node. These subgraphs are fed as features into a graph convolutional neural network (appropriately designed) that predicts any unknown rating.

The paper is overall well written. The approach (of using subgraph connectivity patterns as features) is clever, novel (at least in my knowledge) and neatly sidesteps the need for extra metadata/features. The numerical results are also thorough and compelling.

Minor comments:
- Isn't Alg 1 just ordinary bread-first search? Perhaps I missed something.
- In Table 2 why do some methods have error bars and others don't?
- Fig 3 is not very informative.




**Experience Assessment:**

I have published in this field for several years.

**Review Assessment: Checking Correctness Of Derivations And Theory:**

I assessed the sensibility of the derivations and theory.

**Review Assessment: Checking Correctness Of Experiments:**

I assessed the sensibility of the experiments.

**Review Assessment: Thoroughness In Paper Reading:**

I made a quick assessment of this paper.

---

> ### Author Response · Authors · 2019-11-11
> **Response to Reviewer 1**
>
> Thank you very much for your comments and affirmations!
>
> 1) Isn't Alg 1 just ordinary bread-first search? Perhaps I missed something.
> Yes, Algorithm 1 uses BFS to extend nodes, and extracts the enclosing subgraph induced by these nodes from the large bipartite graph.  We include Algorithm 1 for preciseness and completeness.
>
> 2) In Table 2 why do some methods have error bars and others don't?
> Those with error bars were ran by ourselves, and those without error bars were taken from their original papers (no error bars reported in the original papers).
>
> 3) Fig 3 is not very informative.
> We will try to render better visualizations.

---

### Official Review · AnonReviewer3 · 2019-10-27
**Official Blind Review #3**

**Rating:** 6

**Review:**

This paper presents an inductive matrix completion model using graph neural networks. The proposed method assumes the rating between a user-item pair is determined by this pair's surrounding sub-graph via a neural network. It claims to be an inductive model and don't need any side information as it only uses the surrounding sub-graph structure to give predictions. The experiments also show the promising results on matrix completion tasks, sparse rating matrix cases and transfer learning tasks

The paper is well written and easy to follow. I think the "cold-start" setting this method is mainly focusing on is worth exploring. In this setting, new users have given a few ratings but the quality of their side information is still low. It seems to me that this setting is common in real case. For example, newly registered Netflix users may quickly watch some videos without completing their profile information. However, when the "new" users have given some ratings, we can re-train a traditional MC model including the new ratings. But I believe the proposed method of this paper will be useful when re-training cannot be done very frequently.

I have the following concerns or questions.

1. The paper claims the proposed method doesn't need any side information. However, it seems that the side information can be integrated into the model such as concatenating with the one-hot encoding. Have you tried to use the side information for IGMC?

2. The claims in Section 4 don't make much sense to me. For example, it doesn't seem to me that GC-MC can't distinguish Figure (a) and (b) because GC-MC will take the whole graph into consideration. Also, why will node-based approaches push all nodes to have similar embeddings with multiple layers? Could you give some experimental verifications?

3. It's unclear to me how Eq. (6) helps the performance. It will be better to compare a version without using L_{ARR}. Moreover, I believe Eq. (6) can be designed better.

4. GC-MC uses bilinear decoder to give probability predictions for different ratings. Is there any particular reason why you choose squared loss as in Eq. (5) instead of a bilinear decoder?

5. I have a big concern for the scalability. If we want to make recommendations for a user from a million movies, the proposed method seems to need to compute Eq. (4) for one million times. Is there any approximate way to speedup it?

Overall, I think the proposed method is interesting and the experimental results are impressive.


**Experience Assessment:**

I have published one or two papers in this area.

**Review Assessment: Checking Correctness Of Derivations And Theory:**

N/A

**Review Assessment: Checking Correctness Of Experiments:**

I assessed the sensibility of the experiments.

**Review Assessment: Thoroughness In Paper Reading:**

I read the paper at least twice and used my best judgement in assessing the paper.

---

> ### Author Response · Authors · 2019-11-11
> **Response to Reviewer 3**
>
> Thank you very much for your insightful comments.
>
> 1) Have you tried to use the side information for IGMC?
> Yes, initially we have tried to use side information for IGMC too, and it indeed improved the performance (ML-100K 0.910 -> 0.907 (side)). But later after we improved IGMC to reach 0.905 RMSE on ML-100K using adjacent rating regularization, we found that side information is no longer helpful. This is somewhat expected: GC-MC reaches 0.905 on ML-100K with the help of side information, while IGMC reaches 0.905 without using side information. We hypothesize that the benefits of side information reduce with the better modeling of graph structure features.
>
> 2) The claims in Section 4 don't make much sense to me. For example, it doesn't seem to me that GC-MC can't distinguish Figure (a) and (b) because GC-MC will take the whole graph into consideration. Also, why will node-based approaches push all nodes to have similar embeddings with multiple layers? Could you give some experimental verifications?
> By claiming that GC-MC cannot discriminate figures (a) and (b) we mean GC-MC cannot discriminate them merely from the one-hop neighborhood of the two target nodes. Given multiple graph convolution layers, GC-MC will be able to eventually discriminate them by using the whole graph information. However, stacking many graph convolution layers might push all node embeddings to be similar, which is known as over-smoothing [1]. Over-smoothing is deeply caused by GCN’s repeated smoothing effect which makes node embeddings in the same cluster similar so that semi-supervised node classification becomes easier. We indeed observed this effect in our experiments with PinSage: the RMSE on ML-100K using 1, 2, 3, 4 graph convolution layers were: 0.951, 1.058, 1.162, 1.272, respectively, which indicates that increasing GCN depth makes it harder and harder to discriminate nodes and predict ratings.
>
> 3) It's unclear to me how Eq. (6) helps the performance. It will be better to compare a version without using L_{ARR}. Moreover, I believe Eq. (6) can be designed better.
> ARR helps the performance because it takes into consideration the rating magnitude information instead of just treating ratings as different classes. By encouraging adjacent ratings (e.g., 4 and 5) to have similar parameter matrices, adjacent ratings will pass more similar messages to center during message passing, thus indicating similar preferences from nodes. For example, suppose user A and item B are connected by a rating 5, and user A and item C are connected by a rating 4. We know A has a high preference for both B and C. With ARR, the graph convolution tends to pass similar messages from A to  B and C since ratings 4 and 5 have similar parameter matrices, which effectively conveys user A’s similar preference for B and C.
> We show some numerical results in Appendix A of the revised paper. IGMC’s RMSE on ML-100K without ARR is 0.908, and using ARR improves it to 0.905. Note that this seemingly small improvement is actually very impressive due to the already low RMSE 0.908 without ARR. We agree that ARR is only one possible way to consider rating magnitude, and is able to be designed better. We leave it for future work.
>
> 4) GC-MC uses bilinear decoder to give probability predictions for different ratings. Is there any particular reason why you choose squared loss as in Eq. (5) instead of a bilinear decoder?
> We choose to model ratings as continuous numbers instead of different classes like GC-MC does. The main reason is that we want to model the continuity of ratings explicitly. It is also why we introduced adjacent rating regularization (ARR) to take into consideration of rating magnitude (a rating of 5 is similar to a rating of 4, but vastly different from a rating of 1).
>
> 5) I have a big concern for the scalability. If we want to make recommendations for a user from a million movies, the proposed method seems to need to compute Eq. (4) for one million times. Is there any approximate way to speedup it?
> Thanks for the great question. We tested IGMC’s inference time on ML-100K. It took 62s to make predictions for all 20,000 testing subgraphs using a batch size of 50 on a single CPU+GPU, which means 1 million movies take around 50 minutes. Thus, IGMC is perhaps not suitable for recommending millions of items. One workaround is multi-stage recommendation, which firstly uses simpler methods to roughly select top 1000 items from 1 million, and then uses IGMC to make final accurate predictions. Predicting 1000 items only takes 3s. There are also many possible ways to accelerate IGMC, such as using less GCN layers, decreasing subgraph sizes, increasing batch sizes, using less channels, etc.
>
> [1] Li, Qimai, Zhichao Han, and Xiao-Ming Wu. "Deeper insights into graph convolutional networks for semi-supervised learning." Thirty-Second AAAI Conference on Artificial Intelligence. 2018.

---

### Decision · Program_Chairs · 2019-12-19

**Decision:**

Accept (Spotlight)

**Comment:**

This paper proposes a novel technique for matrix completion, using graphical neighborhood structure to side-step the need for any side-information.

Post-rebuttal, the reviewers converged on a unanimous decision to accept. The authors are encouraged to review to address reviewer comments.